# Evaluation of Macular Ganglion Cell-Inner Plexiform Layer in Children with Deprivational Amblyopia Who Underwent Unilateral Cataract Surgery

**DOI:** 10.3390/medicina59010013

**Published:** 2022-12-21

**Authors:** Marta Świerczyńska, Agnieszka Tronina, Bartosz Filipek-Janiszewski, Erita Filipek

**Affiliations:** 1Department of Ophthalmology, Faculty of Medical Sciences, Medical University of Silesia, Kornel Gibiński University Clinical Center, 40-055 Katowice, Poland; 2Department of Paediatric Ophthalmology, Faculty of Medical Sciences, Medical University of Silesia, Kornel Gibiński University Clinical Center, 40-055 Katowice, Poland; 3The Faculty of Medicine, Medical University of Gdańsk, M. Skłodowskiej-Curie 3a Street, 80-210 Gdańsk, Poland

**Keywords:** cataract surgery, pediatric cataract, unilateral cataract, macular GCIPL thickness, ganglion cell layer, inner plexiform layer, deprivational amblyopia

## Abstract

*Background and Objectives:* The aim of the study is to assess macular ganglion cell and inner plexiform layer (mGCIPL) thickness in deprivational amblyopic eyes (AE), fellow non-amblyopic eyes (FE) and normal eyes (NE) using spectral. domain optical coherence tomography (SD-OCT). *Materials and Methods:* Twenty two children (64% boys) who underwent surgical removal of unilateral congenital or developmental cataracts and exhibited visual impairment despite postoperative visual rehabilitation were included in the study. Cataract surgery was performed in patients aged 55.82 ± 35.85 months (range 6 to 114 months). The mean age of the study group was 9.73 ± 2.85 years (range 5 to 15 years). The comparison group consisted of 22 healthy age- and gender-matched children. The best corrected visual acuity (BCVA) after surgery was: 0.75 ± 0.27 (range 0.3 to 1.3) in AE, 0.1 ± 0.13 (range 0 to 0.5) in FE and 0.04 ± 0.07 (range 0 to 0.2) in NE. OCT scans were performed in all patients and subsequently corrected for axial length related magnification errors. *Results:* The average thickness of mGCIPL was 70.6 ± 11.28 μm in AE; 77.50 ± 6.72 μm in FE and 81.73 ± 5.18 μm in NE. We found that mGCIPL was statistically significantly thinner in deprivation AE compared to FE (*p* = 0.038) and NE (*p* = 0.0005). The minimum thickness of mGCIPL was respectively: 62.68 ± 13.2 μm, 70.3 ± 7.61 μm, and 74.5 ± 5.47, and also differed between AE and FE (*p* = 0.023) and AE and NE (*p* = 0.0004). Also, measurements in the inferior, inferotemporal, and superotemporal sectors showed thinning of mGCIPL in AE compared to NE. *Conclusions:* This analysis may suggest that deprivational amblyopia caused by unilateral congenital or developmental cataract in children may be associated with mGCIPL thinning.

## 1. Introduction

Ambylopia, after myopia, is the second most common cause of reduced visual acuity in children, affecting about 1.6–3.6% of the population. Among its most common causes are anisometropia, strabismus, binocular high refractive defects, visual deprivation (congenital cataract, ptosis, lack of corneal transparency) or the co-occurrence of any of the above [1]. On the other hand, cataracts are considered one of the leading causes of blindness and visual deprivation in children worldwide. Opacity of the natural lens prevents visual stimulation during the development of the neuronal network between the retina and visual cortex, which may be associated with irreversible changes [2]. Although screening and early removal of cataracts with replacement of an intraocular lens plays a key role in avoiding deprivational amblyopia [3], postoperative best-corrected visual acuity (BCVA), especially among children with unilateral cataracts, is sometimes difficult to predict [4,5].

Initially, the decrease in VA was considered to be primarily related to changes involving the visual cortex and lateral geniculate nucleus [6,7,8], but studies in animals with unilateral lid closure or dark rearing have also shown the occurrence of abnormalities affecting retinal microstructures [9,10,11,12,13,14]. Impaired functioning of retinal ganglion cells (RGCs) assessed by pattern electroretinography (PERG) has also been observed in humans [15,16]. On the other hand, with the development of optical coherence tomography (OCT), correlations between vision loss and peripapillary retinal nerve fiber layer (RNFL), macular thickness, changes in retinal layering or retinal vascular structure became apparent. Nevertheless, the results of presented studies often remain contradictory [17,18,19,20,21,22,23,24,25,26,27,28,29,30]. In addition, there are few works evaluating changes in retinal layer structure under the influence of sensory deprivation after congenital or developmental cataract surgery [17,18,20,22,26]. Considering that light deprivation has the strongest impact on the formation of visual impairment [31,32], further research on this issue may bring new insights into understanding the process of visual impairment.

The purpose of the present study is to examine whether the thickness of the macular ganglion cell-inner plexiform layer (mGCIPL), assessed by spectral-domain OCT (SD-OCT) using correction for axial length (AL)-related magnification errors, differs between deprivational amblyopic eyes, after the removal of unilateral congenital or developmental cataracts, fellow non-amblyopic eyes, and normal eyes from the control group.

## 2. Methods

### 2.1. Study Population

Among children who underwent unilateral congenital or developmental cataract surgery at the Department of Pediatric Ophthalmology at the University Clinical Center in Katowice, Poland, 22 patients were included in the analysis in whom, after a minimum of 4 years after surgery and despite the treatment of vision impairment (correction of refractive error, penalization treatment), a diagnosis of vision impairment of the operated eye was made. The control group consisted of 22 age- and sex-matched, healthy children, routinely examined at the Children’s Outpatient Ophthalmology Clinic. The study was conducted in accordance with the principles of the Declaration of Helsinki. Informed consent for the study was obtained from the parents or legal guardians, and additionally from the patients themselves, as they were able to understand the purpose of the research being performed.

### 2.2. Inclusion Criteria

Inclusion criteria for the study group: (1) status after unilateral congenital or developmental cataract surgery; (2) lens opacity that obscured the fundus insight in mydriasis; (3) cataract surgery performed at age ≤ 10 years; (4) uncomplicated cataract surgery and postoperative period; (5) visual impairment of the operated eye (visual impairment was determined when the difference in BCVA between the eyes was ≥2 rows on the Snellen chart in the absence of an organic cause; (6) visual impairment of the operated eye despite the use of visual impairment therapy (refractive correction, penalization methods of the better-seeing eye) for ≥4 years according to the ophthalmologist’s recommendations; (7) obtaining a good quality OCT scan (OCT scan signal strength ≥ 6 was considered acceptable, maximum OCT scan signal strength = 10).

### 2.3. Exclusion Criteria

Patients with the following were excluded from the study: (1) birth prior to completion of gestational week 37 and/or birth weight < 2500 g; (2) pre- or post-operative glaucoma (3) other ophthalmic disorders including: strabismus, nystagmus, microphthalmia, micro- or megalocornea, keratoconus, coloboma, aniridia, optic nerve hypoplasia, chronic uveitis, persistent hyperplastic primary vitreous body, other anterior or posterior eye disorders; (4) traumatic or complicated cataracts; (5) family history of congenital cataracts; (6) appearance of complications during cataract surgery and/or postoperative complications (7) history of ocular or head trauma; (8) previous ophthalmic surgery; (9) occurrence of congenital systemic infections, exposure to toxins; (10) neurological comorbidities, developmental delay; (11) anisometropia ≥ 3.0 diopters; (12) lack of patient cooperation during the examination, segmentation errors of individual layers, poor quality of the OCT scans (OCT scan signal strength <6).

### 2.4. Intraoperative Procedures

One experienced surgeon (E.F.) performed all cataract removal surgeries. Patients were operated under general anesthesia. The anterior chamber was opened by lateral paracentesis and a tunnel cut (1 mm) in the transparent cornea. Provisc viscoelastic material (Alcon Laboratories Inc., TX, USA) was injected into the anterior chamber to maintain depth. Anterior capsulorhexis was performed using an electric capsulotome (Erbe USA Inc., GA, USA). After hydrodissection, the lens masses were removed using a two-handed technique with separate aspiration and irrigation (Infiniti Vision System, Alcon Laboratories Inc., TX, USA). Posterior capsulorhexis and anterior vitrectomy were performed to avoid opacification of the visual axis in children younger than 5 years. Viscoelastic material was injected into the anterior chamber and lens capsule, the tunnel incision was widened to 2.8 mm and an artificial lens was implanted using an injector. The viscoelastic was then removed, and a 10-0 Nylon safety suture was placed over the tunnel cut, and the side port was sealed with a balanced salt solution (BSS).

### 2.5. Examination

In all patients, the following were assessed: BCVA (using Snellen chart; BCVA value was converted to logarithm of minimum angle of resolution (LogMAR) for statistical analysis), ocular motility, intraocular pressure (using Goldmann applanation or non contact tonometry). In addition, a slit-lamp examination of the anterior segment of the eye, refractive measurements (recorded as spherical equivalent (SE)—autorefractometer KR-800, Topcon, Tokyo, Japan) and evaluation of the fundus after obtaining pharmacological mydriasis (after a 3-time application of 1% Tropicamide) were performed. AL was measured using an optical biometer (IOLMaster 500, Carl-Zeiss Meditec, Dublin, CA, USA). After pupil dilation, OCT macular scans (OCT Cirrus 6000, Carl-Zeiss Meditec, Dublin, CA, USA) of 512 × 128 were taken and analyzed by GCA algorithm. The program automatically recognized the outer boundaries of the RNFL and the inner boundaries of the IPL. The mGCIPL measurement included minimum, average and sectoral (superior, superonasal, superotemporal, inferior, inferonasal and inferotemporal) thickness (Figure 1). Signal strength ≥ 6 (maximum = 10) was considered acceptable. All examinations were performed by a single physician, uninformed as to which of the study eyes was visually impaired. For children in the control group, the results obtained during the examination of the dominant eye were included in the analysis.

In order to correct AL-related ocular magnification, the Littmann formula (t = p∙q∙s) was used, which was modified by Bennet and later adopted by Kang et al. [33,34,35]. In the equation, t specifies the actual size of the fundus, p is the magnification ratio of the camera, q is the magnification ratio for the eye, and s is the measurement obtained during the OCT examination. The Cirrus HD-OCT was used in this study, for which p is 3.382, and q can be calculated using the formula q = 0.01306 ∙ (AL − 1.82). Considering that the Littmann formula is applicable more for linear than surface magnification, this formula was modified as suggested by Kim et al. [17].

### 2.6. Statistical Analysis

Statistical analysis was performed using STATISTICA 13.3 software (TIBCO Software Inc., Palo Alto, CA, USA). The Shapiro-Wilk test was used to assess the occurrence of normal distribution among the studied variables. For comparison of variables between the amblyopic and the fellow non-amblyopic eyes, the paired Student’s *t*-test or Wilcoxon signed-rank test were used depending on the distribution of the data. For comparison of variables between the amblyopic and age-matched normal eyes, the Student’s *t*-test or Mann-Whitney test were performed according to the data normality. Chi square test was used to assess the difference between groups in terms of gender. A *p*-value < 0.05 was considered to be the level of statistical significance.

## 3. Results

The study included: 22 pseudophakic children (14 boys) aged 5 to 15 years, mean age 9.73 ± 2.85 years, and 22 healthy children (12 boys) aged 5 to 14 years, mean age 9.36 ± 2.46 years. The mean age of patients who underwent surgery was 55.82 ± 35.85 months (range 6 to 114 months). OCT examination was performed 63.0 ± 21.32 months after surgery (range 48 to 132 months). BCVA in each group was: 0.75 ± 0.27 (range 0.3 to 1.3), 0.1 ± 0.13 (range 0 to 0.5), 0.04 ± 0.1 (range 0 to 0.2). The mean SE value was −1.51 ± 2.29 D (range −5.25 to 3.5) in the deprivational amblyopic eyes; −1.73 ± 2.22 in fellow non-amblyopic eyes (range −5.0 to 3.75) and −1.86 ± 2.79 in normal eyes (range −6.0 to 3.5). There were no statistically significant differences between the groups with respect to SE, AL, or signal strength of the tests performed. (Table 1).

The mean thickness of mGCIPL was 70.68 ± 11.28 μm in deprivational amblyopic eyes, 77.50 ± 6.72 μm in fellow non-amblyopic eyes, and 81.73 ± 5.18 μm in normal eyes. We found that mGCIPL was significantly thinner in deprivational amblyopic compared to fellow non-amblyopic eyes (*p* = 0.038) and normal eyes (*p* = 0.0005). The mean thickness of mGCIPL in non-amblyopic was lower than in the control group, but there was no significant statistical difference between the groups (*p* = 0.064).

The minimum thickness of mGCIPL was 62.68 ± 13.20 μm in deprivational amblyopic eyes, 70.36 ± 7.61 μm in fellow non-amblyopic eyes, and 74.5 ± 5.47 μm in normal eyes. There was a significant statistical difference between deprivational amblyopic and fellow non-amblyopic eyes (*p* = 0.023) and normal eyes (*p* = 0.0004). The mean thickness of mGCIPL in fellow non-amblyopic eyes was lower than in the control group, and this was a statistically significant difference (*p* = 0.045). Also, measurements in the inferior, inferotemporal, superotemporal sectors showed differences between deprivational amblyopic and normal eyes (Table 2).

## 4. Discussion

The obtained data indicate that the thickness of mGCIPL in ambylopic eyes is significantly lower compared to fellow non-amblyopic eyes and the control group. Animal studies showed that visual deprivation can cause changes in the structure of the retina, including: degeneration of RGCs [9,10], reduction in nucleolar volume and cytoplasmic crosssectional area of RGCs [11], increase in the number of amacrine synapses in the IPL [12,13], reduction in the number of bipolar synapses in the IPL [12], thinning of the IPL [11,14], and reduction in the density of Müller fibers [14].

Moreover, using OCT angiography (OCTA), it was shown that people, regardless of the type of visual impairment, had significantly lower vessel density in the superficial capillary plexus (SCP) and deep capillary plexus (DCP) compared to the control group. In addition, in amblyopic eyes, the decrease in vascular system density was positively correlated with a decrease in GCL volume, which was significantly lower compared to non-amblyopic fellow eyes and the control group. However, it was not determined whether changes in GCL or CP should be considered as initiating. It has been speculated that a reduction in metabolism due to a decrease in visual stimuli reaching the visually impaired eye may be responsible for the above differences [18]. Can [19], on the basis of her analysis, speculates that it is the reduced number as well as size of RGCs that affects microvascular density. In contrast, Nishikawa et al. [20] confirm a reduction in macular vessel density, but with an increase in the thickness of inner retinal layers comparing amblyopic, fellow non-amblyopic and normal eyes, even after image magnification correction. In addition, the authors attribute the reduction in vessel density to the disturbance or delay that occurs during foveal development.

Zhang et al. [21] demonstrated that in patients with unilateral congenital cataracts, the vascular density of the macular area and optic disc was significantly lower than in normal eyes. In contrast, improved vascular density was noted after cataract surgery. It should be noted that our study included patients who, after unilateral cataract surgery, and after a minimum of 4 years of using vision therapy, still presented vision loss of the operated eye. This may prove that the changes that occurred in the mGCIPL under the influence of sensory deprivation are not always easily reversible. On the other hand, there is a possibility that disorders that were already present before surgery, but went undetected, may have been responsible for the lack of expected VA improvement.

There is limited work regarding the structural changes of the retina that potentially may occur as a result of sensory deprivational amblyopia in children with congenital or developmental cataracts. Results from studies on central macular thickness (CMT) after cataract surgery in children are conflicting [23,24,25]. Hansen et al. [26] showed no significant differences regarding GCL thickness in children after unilateral cataract surgery, but the second, unoperated eye was used as the control group in the before mentioned study. On the other hand, analysis of the patients’ results after bilateral cataract surgery showed that the GCL thickness was greater in the better-seeing eye than in the worse-seeing eye, but the difference was not statistically significant. Also, the results we presented show that the average thickness of mGCIPL in fellow non-amblyopic eyes was lower than in eyes of control patients, but this was also not a statistically significant difference. Observations by other authors show that unilateral amblyopia can cause bilateral changes involving the optic nerve [36] contrast sensitivity function [37], macular scotoma [38]. Therefore, fellow eyes should not be used as a control group.

Kim et al. [17] used correction for AL-related magnification errors and showed that mGCIPL thickness was not statistically different between eyes with deprivational ambylopia, after unilateral cataract surgery versus fellow non-amblyopic and normal eyes. Obtaining a different result may be related to the fact that the average age at which cataract removal surgery was performed in our study group was 55.82 ± 35.85 months, which was higher than in the study by Kim et al. in which unilateral cataracts were diagnosed in patients aged 32.39 ± 25.04, and the time from diagnosis to surgery was 3.87 ± 4.13 months. On the other hand, the BCVA obtained after surgery in the group we analyzed was lower than among the children studied by Kim et al. (0.75 ± 0.27 vs. 0.63 ± 0.51). It is believed that retinal synaptic pathway and retinogeniculate synapses undergo postnatal remodeling under the influence of visual stimulation [39,40]. Moreover, histological and in vivo OCT studies indicate that the development of the fovea continues into the middle teenage years and not just until age five, as previously thought [41]. Thus, it can be speculated that a longer period of sensory deprivation has a greater effect on changes in retinal structure. Further potential reasons for the different results include the difficulty of controlling for depth of visual impairment, duration of visual deprivation, adherence to visual impairment therapy of the study group, or ethnic differences. In addition, patients with aphakic correction period were not evaluated in our analysis.

A study by Park et al. [27], which measured the thickness of each retinal layer, found significant GCIPL thinning in eyes with various types of visual impairment (strabismus, aniso-astigmatism, unilateral ptosis). However, the thickness of the layers was measured manually using SD-OCT calipers. In addition, it was shown that some of the layers examined were thicker in certain areas of the macula and thinner in others, which may suggest that different pathological changes occur in different areas of the macula. Xia et al. [28] analyzed anisometropic amblyopia and found that the thickness of 2 of the 3 GCL quadrants and 3 of the 4 IPL quadrants in the peripheral macular area studied were significantly reduced compared to the control group.

Tugcu et al. [29] showed a significant reduction in GCC thickness in strabismic amblyopia compared to non-amblyopic eyes. In contrast, there was an increase in GCC in the anisometropic and combined group (anisometropia and strabismus) in amblyopic and non-amblyopic eyes compared to the control group. On the contrary, Guagliano et al. [30] detected no differences between GCC thickness in amblyopic, fellow and control eyes. However, it should be kept in mind that the GCC includes the GCL, IPL and RNFL, and studies on the thickness of the latter in visually impaired eyes also remain contradictory [17,20,22,23,26,27,28,29,30], making it difficult to relate the increased data to our results.

The variability in the results presented in the literature may be due to limited study size, differences in age, ethnicity, timing of surgery in the study groups, often lack of comparison to a healthy control group, errors related to manual measurement, use of different versions of the OCT scanning algorithm, or omission of correction of the ocular magnification errors related to refraction and AL. In addition, some of the studies were conducted on visually impaired eyes without considering the cause of the visual impairment.

## 5. Conclusions

Despite the relatively small size of the study group, our analysis may indicate that deprivational amblyopia caused by unilateral congenital or developmental cataract in children may be associated with mGCIPL thinning. However, only a prospective study with a larger study group can confirm our results and determine their clinical significance. In addition, the evaluation of more peripheral areas of the retina could contribute to a better understanding of the reasons for poor visual outcomes in children with unilateral cataracts, regardless of the performance of cataract removal surgery without intra- and postoperative complications.

## Figures and Tables

**Figure 1 medicina-59-00013-f001:**
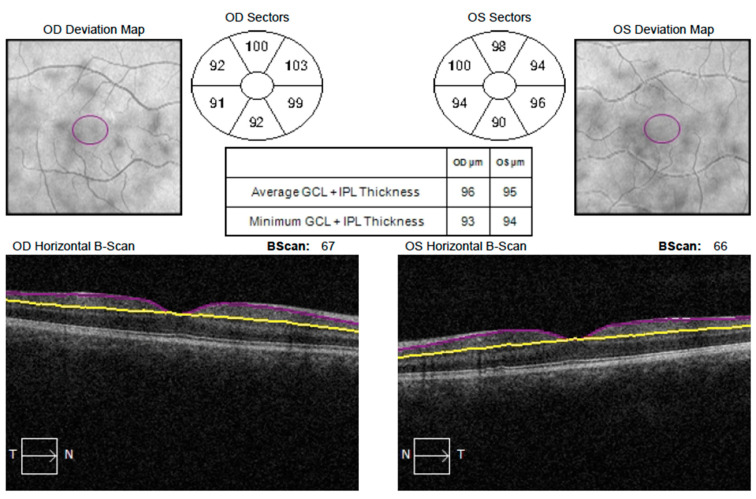
Example of mGCIPL measurement with Cirrus 6000. Visible subdivision into sectors: superior, superonasal, superotemporal, inferior, inferonasal and inferotemporal.

**Table 1 medicina-59-00013-t001:** Demographic and clinical description of patients with deprivational ambylopia and children from control group.

	Deprivational Amblyopic Eyes (22 Eyes) (Mean ± SD)	Fellow Non-Amblyopic Eyes (22 Eyes) (mean ± SD)	Normal Eyes (22 Eyes) (Mean ± SD)	*p*1-Value †	*p*2-Value ‡
Age (years)	9.73 ± 2.85		9.36 ± 2.46	-	0.653
Male (*n* %)	14 (63.6)		12 (54.5)	-	0.540 #
Vision (LogMAR)	0.75 ± 0.27	0.1 ± 0.13	0.04 ± 0.07	0.000 *§	0.000 *¦
SE (D)	−1.51 ± 2.29	−1.73 ± 2.22	−1.86 ± 2.79	0.747	0.646
Axial length (mm)	23.89 ± 1.7	24.18 ± 1.77	24.54 ± 2.08	0.599	0.272
OCT scan signal strength	0.8 ± 0.1	0.81 ± 0.11	0.83 ± 0.13	0.916	0.496

D: diopters; LogMAR: logarithm of the minimum angle of resolution; SD: standard deviation; SE: spherical equivalence; * statistically significant difference (*p* < 0.05); # Comparison was performed using chi square test. † Comparison was performed using a paired *t*-test between amblyopic eyes and fellow non-amblyopic eyes. § Comparison was performed using a Wilcoxon signed-rank test between amblyopic eyes and fellow non-amblyopic eyes. ‡ Comparison was performed using an unpaired *t*-test between amblyopic eyes and age-matched normal eyes. ¦ Comparison was performed using Mann-Whitney test between amblyopic eyes and age-matched normal eyes.

**Table 2 medicina-59-00013-t002:** Comparison of the mGCIPL thickness between deprivational amblyopic eyes, fellow non-amblyopic eyes and age-matched eyes from control group.

GCIPL Thickness (μm)	Deprivational Amblyopic Eyes (22 Eyes) (Mean ± SD)	Fellow Non-Amblyopic Eyes (22 Eyes) (Mean ± SD)	Normal Eyes (22 Eyes) (Mean ± SD)	*p*1-Value †	*p*2-Value ‡
Average	70.68 ± 11.28	77.50 ± 6.72	81.73 ± 5.18	0.038 *§	0.0005 *¦
Minimum	62.68 ± 13.20	70.36 ± 7.61	74.50 ± 5.47	0.023 *	0.0004 *
Superior	78.68 ± 10.06	80.45 ± 7.76	84.18 ± 5.84	0.869 §	0.103 ¦
Superonasal	76.86 ± 9.89	81.27 ± 7.47	83.09 ± 6.04	0.236 §	0.056 ¦
Inferonasal	74.09 ± 13.38	79.32 ± 7.73	81.64 ± 4.45	0.265 §	0.116 ¦
Inferior	68.23 ± 13.53	73.68 ± 8.87	78.36 ± 5.77	0.173 §	0.008 *¦
Inferotemporal	70.54 ± 11.33	76.86 ± 7.45	79.82 ± 5.29	0.025 *§	0.003 *¦
Superotemporal	69.59 ± 11.56	77.36 ± 8.64	79.73 ± 6.07	0.007 *§	0.001 *¦

mGCIPL: macular ganglion cell-inner plexiform layer; SD: standard deviation; * statistically significant difference (*p* < 0.05); † Comparison was performed using a paired *t*-test between amblyopic eyes and fellow non-amblyopic eyes. § Comparison was performed using a Wilcoxon signed-rank test between amblyopic eyes and fellow non-amblyopic eyes. ‡ Comparison was performed using an unpaired *t*-test between amblyopic eyes and age-matched normal eyes. ¦ Comparison was performed using Mann-Whitney test between amblyopic eyes and age-matched normal eyes.

## Data Availability

The dataset analyzed in this study is available from the corresponding author upon reasonable request.

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
