# Peer review of "Evaluation of Macular Ganglion Cell-Inner Plexiform Layer in Children with Deprivational Amblyopia Who Underwent Unilateral Cataract Surgery"

_medicina, 2022, doi:10.3390/medicina59010013_

Round 1

Reviewer 1 Report

The manuscript Evaluation of macular ganglion cell-inner plexiform layer in children with deprivational amblyopia who underwent unilateral cataract surgery” assessed macular ganglion cell and inner plexiform layer (mGCIPL) thickness in deprivational amblyopic eyes (AE), fellow non-amblyopic eyes (FE) from 22 children under unilateral congenital or developmental cataract and normal eyes.

Major issues:

1.I would recommend the authors to analyze the relationship between visual recovery and mGCIPL thickness.

2.The sample size is too small, and how is the sample size calculation?

Minor issues:

Need to define Scan quality index in the methods section.

Need to clarify the statistical analysis methods below the tables.

Author Response

We sincerely thank you for taking the time to review our manuscript and for your valuable comments. We have highlighted all modifications made in the main text in red.

1. I would recommend the authors to analyze the relationship between visual recovery and mGCIPL thickness.

Thank you for the above suggestion. Assessing the correlation between mGCIPL thickness and visual recovery (understood as the difference between visual acuity before surgery and after surgery and undergoing visual rehabilitation) is not possible in our study, because some of the children were too young to objectively examine their visual acuity before cataract surgery, or even in the short interval after surgery.

On the other hand, Spearman's correlation coefficient between visual acuity (VA), after cataract surgery with visual rehabilitation, and average and minimum mGCIPL thickness was not statistically significant (p-value was 0.72 and 0.76, respectively). The lack of correlation between the above values may be due to the fact that the study conducted is retrospective, thus it was impossible to collect accurate data on what the process of visual rehabilitation and compliance of patients and their caregivers looked like, which perhaps prevented the appropriate selection of the study group in the context of assessing functional outcomes. Therefore, we believe that the inclusion of the results of this correlation is not valid in the context of our analysis.

2. The sample size is too small, and how is the sample size calculation?

Epidemiological data on the prevalence of unilateral cataracts in children, 

as well as the visual impairment that occurs as a result of it, are quite limited.

Based on data obtained in the British population [1, 2], we took the liberty of assuming that there are 100 children with unilateral cataracts per 1 million live births, of whom 40 patients will develop visual impairment despite surgery and visual rehabilitation. In Poland, there were just over 6 million live-born children between 2002 and 2014, which allows us to assume that about 600 of them had unilateral cataracts, and 240 may develop visual impairment, accounting for 0.004% of the study population.

Using the sampling calculator, assuming a population size of 6 million; a fraction size of 0.1; a confidence level of 95%, and a maximum error of 10%, the proposed sample size is 35. In contrast, assuming a maximum error of 12.5%, the sample size should be 22.

However, it should be borne in mind that the exclusion criteria used in our study-including prematurity, other concomitant ophthalmic complications, previous ophthalmic surgery, the presence of neurological disorders or developmental delays-make the fraction size even smaller.

In addition, previous studies on unilateral cataracts in children and visual impairment and/or OCT studies have equally small sample sizes [3 - 7]. 

  1. Chan WH, Biswas S, Ashworth JL, Lloyd IC. Congenital and infantile cataract: aetiology and management. Eur J Pediatr. 2012 Apr;171(4):625-30. doi: 10.1007/s00431-012-1700-1

  1. Rahi JS, Dezateux C; British Congenital Cataract Interest Group. Measuring and interpreting the incidence of congenital ocular anomalies: lessons from a national study of congenital cataract in the UK. Invest Ophthalmol Vis Sci. 2001 Jun;42(7):1444-8

  1. Kim YW, Kim SJ, Yu YS. Spectral-domain optical coherence tomography analysis in deprivational amblyopia: a pilot study with unilateral pediatric cataract patients. Graefes Arch Clin Exp Ophthalmol. 2013 Dec;251(12):2811-9. doi: 10.1007/s00417-013-2494-1 (14 patients)

  1. Park KA, Park DY, Oh SY. Analysis of spectral-domain optical coherence tomography measurements in amblyopia: a pilot study. Br J Ophthalmol. 2011 Dec;95(12):1700-6. doi: 10.1136/bjo.2010.192765. (20 patients)

  1. Bansal P, Ram J, Sukhija J, Singh R, Gupta A. Retinal Nerve Fiber Layer and Macular Thickness Measurements in Children After Cataract Surgery Compared With Age-Matched Controls. Am J Ophthalmol. 2016 Jun;166:126-132. doi: 10.1016/j.ajo.2016.03.041. (15 patients)

  1. Nishikawa N, Chua J, Kawaguchi Y, Ro-Mase T, Schmetterer L, Yanagi Y, Yoshida A. Macular Microvasculature and Associated Retinal Layer Thickness in Pediatric Amblyopia: Magnification-Corrected Analyses. Invest Ophthalmol Vis Sci. 2021 Mar 1;62(3):39. doi: 10.1167/iovs.62.3.39. (22 patients)
  2. Wang J, Smith HA, Donaldson DL, Haider KM, Roberts GJ, Sprunger DT, Neely DE, Plager DA. Macular structural characteristics in children with congenital and developmental cataracts. J AAPOS. 2014 Oct;18(5):417-22. doi: 10.1016/j.jaapos.2014.05.008. (22 patients)

4. Need to define "Scan quality index" in the method section.

The term "Scan quality index" is equivalent to quality of the OCT scan and signal strength. These terms are standardized in section 2.2 Inclusion criteria, 2.3 Exclusion criteria and Table 1.

5. Need to clarify the statistical analysis methods below the tables.

Clarification of the tests used is introduced in Section 2.6 Statastical Analysis and under each of the tables with results.

Reviewer 2 Report

Great job on the design and methodology of the study.  I would like to put forth some comments and suggestions:

1. How was the sample size determined for the study? 

2. During statistical analysis were any modification made to the t test to account for multiple comparisons? 

3. In the results section, authors mention: the mean age of patients who underwent surgery was 55.82+/-35.85 months (range: 6 to 114 months). 114 months equals to approximately 9 years, however in the inclusion criteria authors mention including only children when surgery was done at <7 years of age. If authors could clarify this and it would be helpful to provide the age distribution of the cohort as well.

4. Was the control group, matched for refractive error as well? I find it very surprising that there was no difference in the refractive error of the operated eye and the normal population. Were they not not under corrected for the IOL power when implanted at <7 years of age? 

5. If the authors could consolidate the discussion more, perhaps  discussing prior studies in a table , it would make it more comprehensible.

Author Response

We sincerely thank you for taking the time to review our manuscript and for your valuable comments. We have highlighted all modifications made in the main text in red.

  1. How was the sample size determined for the study?

Epidemiological data on the prevalence of unilateral cataracts in children, 

as well as the visual impairment that occurs as a result of it, are quite limited.

Based on data obtained in the British population [1, 2], we took the liberty of assuming that there are 100 children with unilateral cataracts per 1 million live births, of whom 40 patients will develop visual impairment despite surgery and visual rehabilitation. There were just over 6 million live-born children in Poland between 2002 and 2014, which suggests that about 600 of them had unilateral cataracts, and 240 of them may develop visual impairment, accounting for 0.004% of the study population.

Using the sampling calculator, assuming a population size of 6 million; a fraction size of 0.1; a confidence level of 95%, and a maximum error of 10%, the proposed sample size is 35. In contrast, assuming a maximum error of 12.5%, the sample size should be 22.

However, it should be borne in mind that the exclusion criteria used in our study-including prematurity, other concomitant ophthalmic complications, previous ophthalmic surgery, the presence of neurological disorders or developmental delays-make the fraction size even smaller.

In addition, previous studies on unilateral cataracts in children and visual impairment and/or OCT studies have equally small sample sizes [3 - 7].

  1. Chan WH, Biswas S, Ashworth JL, Lloyd IC. Congenital and infantile cataract: aetiology and management. Eur J Pediatr. 2012 Apr;171(4):625-30. doi: 10.1007/s00431-012-1700-1

  1. Rahi JS, Dezateux C; British Congenital Cataract Interest Group. Measuring and interpreting the incidence of congenital ocular anomalies: lessons from a national study of congenital cataract in the UK. Invest Ophthalmol Vis Sci. 2001 Jun;42(7):1444-8

  1. Kim YW, Kim SJ, Yu YS. Spectral-domain optical coherence tomography analysis in deprivational amblyopia: a pilot study with unilateral pediatric cataract patients. Graefes Arch Clin Exp Ophthalmol. 2013 Dec;251(12):2811-9. doi: 10.1007/s00417-013-2494-1 (14 patients).

  1. Park KA, Park DY, Oh SY. Analysis of spectral-domain optical coherence tomography measurements in amblyopia: a pilot study. Br J Ophthalmol. 2011 Dec;95(12):1700-6. doi: 10.1136/bjo.2010.192765. (20 patients).

  1. Bansal P, Ram J, Sukhija J, Singh R, Gupta A. Retinal Nerve Fiber Layer and Macular Thickness Measurements in Children After Cataract Surgery Compared With Age-Matched Controls. Am J Ophthalmol. 2016 Jun;166:126-132. doi: 10.1016/j.ajo.2016.03.041. (14 patients).

  1. Nishikawa N, Chua J, Kawaguchi Y, Ro-Mase T, Schmetterer L, Yanagi Y, Yoshida A. Macular Microvasculature and Associated Retinal Layer Thickness in Pediatric Amblyopia: Magnification-Corrected Analyses. Invest Ophthalmol Vis Sci. 2021 Mar 1;62(3):39. doi: 10.1167/iovs.62.3.39. (20 patients).

  1. Wang J, Smith HA, Donaldson DL, Haider KM, Roberts GJ, Sprunger DT, Neely DE, Plager DA. Macular structural characteristics in children with congenital and developmental cataracts. J AAPOS. 2014 Oct;18(5):417-22. doi: 10.1016/j.jaapos.2014.05.008. (20 patients).

2. During statistical analysis were any modifications made to the t test account for multiple comparisons?

For comparison of variables between the amblyopic and the fellow non-amblyopic eyes, the paired Student's t-test or Wilcoxon signed-rank test were used depending on the distribution of the data. For comparison of variables between the amblyopic and age-matched normal eyes, the Student's t-test or Mann-Whitney test were performed according to the data normality.

The specification of the tests used is introduced in section 2.6 Statastical Analysis and under each of the result tables.

3. In the result section, the authors mention: the mean age of patients who underwent surgery was 55.82 +/- 35.85 months (range: 6 to 114). However, in the inclusion criteria authors mention including only children when surgery was done at <7 of age. If authors could clarify this and it would be helpful to provide the age distribution of the cohort as well.

We kindly thank you for pointing out this inaccuracy. The age range of patients who underwent cataract surgery was from 6 to 114 months. However, the inclusion criterion was the age of surgery <10 years old. The change was made in section 2.2 Inclusion criteria. Was the control group matched for refractive error as well? I find it very surprising that there was no difference in the refractive error if the operated eye and the normal population. Were they not under corrected for the IOL power when implanted at <7 years of age?

4. Was the control group matched for refractive error as well? I find it very surprising that there was no difference in the refractive error if the operated eye and the normal population. Were they not under corrected for the IOL power when implanted at <7 years of age?

Gordon et al [1] proposed that suggested intraocular lens (IOL) powers are calculated using the Sanders-Retzlaff-Kraff (SRK) formula, which uses axial eyeball length and keratometry data (if the patient is poorly cooperative, these measurements are taken under general anesthesia before surgery). In contrast, third-generation formulas: SRK/II SRK/T and Hoffer Q provide even better postoperative results [2]. Given that refraction changes as the child grows, and thus the axial length of the eyeball increases, the intent of surgery is to undercorrect in young children to pre-empt decreasing hyperopia and excessive myopic shift. 

Dahan et al [3] suggested reducing implant power by 20% in children <2 years old and 10% less in children >2 years old. They also recommended using an implant power of 28 D for an eye axis length of 17 mm, 27 D for a length of 18 mm, 26 D for 19 mm, 24 D for 20 mm, 22 D for 21 mm. In contrast, Leskul et al. [4] suggest the following IOL adjustments: 30% less in children aged 6 months to 1 year, 25% less at ages 1-2 years, 20% less at ages 2-3 years, 15% less at ages 3-4 years, 10% less at ages 4-5 years, undercorrection by 2 D in children aged 5-8 years and undercorrection by 1 D at ages 8-10 years. 

Initial postoperative hyperopia in young children is targeted and the refractive difference corrected with glasses. However, with the completion of the elongation of the axial length of the eyeball, which stabilizes around age 7-10, the refractive defect also stabilizes. In our analysis, the mean age in the study group was 9.73 ± 2.9 years; therefore, comparison of spherical equivalent (SE) between the study groups showed no statistically significant differences.

1 Gordon RA, Donzis PB. Refractive development of the human eye. Arch Ophthalmol. 1985 Jun;103(6):785-9. doi: 10.1001/archopht.1985.01050060045020. PMID: 4004614.

2 Lee BJ, Lee SM, Kim JH, Yu YS. Predictability of formulas for intraocular lens power calculation according to the age of implantation in paediatric cataract. Br J Ophthalmol. 2019 Jan;103(1):106-111. doi: 10.1136/bjophthalmol-2017-311706.

3 Dahan E, Drusedau MU. Choice of lens and dioptric power in pediatric pseudophakia. J Cataract Refract Surg. 1997;23 Suppl 1:618-23. doi: 10.1016/s0886-3350(97)80043-0. 

4 Lekskul A, Chuephanich P, Charoenkijkajorn C. Long-term outcomes of intended undercorrection intraocular lens implantation in pediatric cataract. Clin Ophthalmol. 2018 Oct 2;12:1905-1911. doi: 10.2147/OPTH.S176057.

5. If the authors could consolidate the discussion more, perhaps discussing prior studies in a table, it would make it more comprehensible.

Thank you very much for the above suggestion. However, due to the variety of studies cited in the discussion (different study groups due to the inclusion and exclusion criteria used, age of children, follow-up period, selection of control group, etc.; type of study, evaluation of various parameters and their measurement methods), we feel that creating a table summarizing studies with similar topics will not benefit the readers.

To date, there has been one study on the issue we are addressing, i.e., an analysis of OCT studies in children with deprivational amblyopia caused by unilateral cataract. This study is described in the discussion, while the other studies with related topics are arranged in a logical sequence.

However, if, in your opinion, the discussion undeniably requires shortening, modification and/or tabulation of the data, we will, of course, try to make corrections.  

Round 2

Reviewer 1 Report

 All the questions have been explained and corrected by the authors, except one. How do you figure OCT scan strength out?from signal strength?It needs to be clarified.

Author Response

Thank you for accepting the modifications we previously sent.

Yes, OCT scan strength was evaluated based on OCT scan signal strength.

This value is evaluated automatically and appears after the scan is performed. OCT scan signal strength ≥ 6 was considered as acceptable.

We have highlighted in red color changes made in the text.

We hope that the above explanation is rewarding.
